# Changing patterns of general practice services during a period of public sector investment in Britain

**Motab Aljohani** [1,2] *****, **Michael Donnelly**[2], **Ciaran O'Neill**[2]

**1** Public Health Department, College of Health Science, Saudi Electronic University, Riyadh, Saudi Arabia,
**2** Centre for Public Health, School of Medicine, Dentistry and Biomedical Sciences, Queen's University Belfast, Belfast, Northern Ireland

***** Maljohani01@qub.ac.uk

## Abstract

### Introduction

Given the importance of GP care to the public's health, it is important that we understand how patterns of service use change as levels of investment change. This study investigated GP use in Britain in conjunction with use of outpatient services during a period of investment and during a period of austerity.

### Method

The study used data from the British Household Panel Survey (BHPS) that included service use, morbidity (as an indicator of need) and socio-demographic characteristics (e.g., employment, age, education, and sex). Data for 2000, 2004, and 2008, were specifically chosen for comparison with data from 2015, 2016 and 2017. Service use and respondent characteristics were described using measures of central tendency and dispersion. Multivariable analyses were undertaken using recursive bivariate probit (RBVP) and probit analyses separately for each study year. All analyses were adjusted for cross-sectional weighting.

### Results

BHPS respondents who used outpatient services or GP services had higher morbidity compared to survey participants who did not. Older people, people with lower educational attainment and employed people had higher mean morbidity indices in each study year as did females. Morbidity among service users tended to decline slightly over time. RBVP analyses revealed a significant positive correlation in residuals between outpatient and GP functions in 2000 and 2004 but not 2008. GP consultations and outpatient use remained largely unrelated to socio-economic factors in each year. Survey participants who reported hearing or vision impairment conditions were consistently less likely to use GP or outpatient services in 2000 and 2004, in 2008.

**Data Availability Statement:** The data used in this study is sourced from the British Household Panel Survey (BHPS), collected by the Institute for Social and Economic Research at the University of Essex and made available by the UK Data Archive. The

datasets are publicly available through the UK Data Service; https://www.understandingsociety.ac.uk/about/british-household-panel-survey.

**Funding:** This work is funded by Saudi Electronic University as part of a PhD. The funders had no role in study design, data collection and analysis, decision to publish, or preparation of the manuscript.

**Competing interests:** None

## Conclusion

The results are broadly indicative of stable relationships in service use during a period of healthcare investment but change during austerity. Those who reported, vision, hearing, and skin conditions were consistently less likely to report use of GP or outpatient services, controlling for other aspects of health.

## Introduction

General practitioners (GPs) are a core element of primary care services and play a key role in the efforts to achieve efficient and equitable delivery of healthcare. The availability of, and access to, GP services contributes to population health [1], effective cost containment [2–4] and the promotion of equity objectives [5–7]. The first and most frequent point of contact between the UK public and the National Health Service tends to be a GP who act as gatekeeper to further medical, diagnostic and specialist services in secondary care [8]. While issues with access have worsened in recent years [9], access remains free at the point of use to all residents [10]. There is a need to improve understanding about the determinants of GP utilisation and how determinants and GP use may change in relation to contextual changes given the central role of GPs to the operation of the healthcare system.

The period 2000–2008 was one of significant reorganization in the UK National Health Service (NHS). With respect to GP services, many GPs began the period having recently been fundholders who in addition to providing primary care, directly commissioned care for their patients from hospital trusts. GPs were reorganized first into primary care groups (PCGs) established in 1999 after GP fundholding was abolished, and then into Primary Care Trusts (PCTs) which took over their commissioning role [11, 12]. In 2004 a new contract was introduced and applied across the UK that changed the responsibilities of GPs as well as introducing an element of performance related pay under the Quality Outcomes Framework (QOF) [12]. QOF influenced how GPs interacted with patients and with secondary care. For example, GPs assumed more responsibility for managing chronic conditions [13]. Indeed, it has been argued the QOF changed the process of care and impacted on outcomes [14].

These changes were implemented within a context of significant investment across the health service including secondary care. Annual spending on public healthcare rose steadily as a percentage of GDP from 5.3% in 1997/98 to 8.26% in 2009, the number of GPs per head of population increased throughout the period, though this increase was not uniform, and indeed there was a fall in the full-time equivalent number of GPs in specific years [1, 12, 15]. The number of GPs per head of population increased between 1997 and 2009 [15] as did their workload—between 1995 and 2008, the number of GP consultations increased by 75% from 171 million to more than 300 million, while the number of consultations per patient per year rose by 11% (3 consolations per-patient-year in 1995 and 1.9 in 2008) [16]. It has been estimated that between 2000 and 2008 the average number of GP consultations per patient increased from 5.3 to 6.8 though face-to-face consultations fell from 3.7 to 3.3 between the two time points [17].

Changes in structure, funding and incentives during this period had the potential to impact on satisfaction and service use [14]. The expansion of services might be expected to result in an expansion of use and impact on onward referral to secondary care. For example, the introduction of the QOF which incentivised management of chronic conditions in primary care may have changed in relative terms the threshold for onward referral of specific conditions

covered by QOF. Equally, a more generous settlement with respect to GP services may have facilitated a narrowing of the gap between need and service provision, for example, with respect to hearing impairment where delayed diagnosis in primary care and onward referral to specialist services had been reported [18]. In turn, this extends to multi-morbidity [19] where conditions included in QOF, within a context of expanded supply may have produced changes in patterns of service use and onward referral. How the combination of these changes affected use is, however, difficult to predict and requires empirical testing. Clearly though, a fuller understanding of GP utilization must make reference to outpatient utilization and consider how users interpret a consultation and how an administrative record categorises a service contact.

GP administrative records may offer insights into patterns of service use and related changes. Studies using these however provide potentially ambiguous results for the period 2000–2008. While, for example, the average number of consultations is reported [17] as rising from 5.3 to 6.8, the number of face to face consultations with a GP are reported to have fallen from 3.7 to 3.3. These sources are mute moreover in terms of public perceptions of changes in use–for example, where some but not all may consider any contact with a GP a consultation, but others only consider a face to face contact one. How patterns of service use changed when considered together with outpatient services also remains unclear given they may be both a complement and a substitute to GP services.

In this paper, we analyse three waves of the British Household Panel Survey (BHPS) and of its successor the understanding society survey to examine the relationship between GP and outpatient service use and a changing funding environment.

## Methods and materials

### Materials

Data were taken from the British Household Panel Survey (BHPS) a nationally representative survey of adults aged 16 and over conducted annually in Britain. As data are anonymised and publicly available from the UK Data Archive at Essex ethics committee approval for their analysis was not required. The survey uses a sampling design with an approximately equal probability of selection method [10]. In addition to questions on the household (income, size, composition) and individual characteristics (age, education, gender, and health conditions), the survey identifies utilization of various health services by the individual, including GP and outpatient use. To allow comparisons of service across multiple years, data were taken from years when questions were asked in a consistent manner and for a time period during which expenditure on care grew sharply; specifically we chose 2000, 2004 and 2008 to examine service use. Questions on service use in BHPS identify frequency of use based on categories. For the purpose of our analyses, however, service use was dichotomised with respect to both GP and outpatient care, taking the value 1 if the respondent had used the service in the preceding 12 months and 0 otherwise. While this resulted in some data loss it still allowed trends in use to be estimated, while avoiding the need to interpolate within categories. It also permitted the use of joint models to estimate the relationship between service use and respondent characteristics within multivariate analyses.

BHPS provides details on an extensive range of possible health conditions as set out in (Tables 1–3). In respect of each, the presence of the condition was captured as 1 if currently experienced by the respondent and 0 otherwise. The conditions were also aggregated to provide an index of morbidity (in fashion consistent with previous studies of service use) [20]. As noted, the survey also provides details a range of socio-demographic characteristics for respondents. These include employment status, sex, marital status, age, education, income and

**Table 1. Descriptive statistics of the study sample 2000.**

| Variable/parameter | Mean (SE) or Proportion (%) |
|---|---|
| GP Visit | 69.75 |
| Outpatient Visit | 24.81 |
| Married | 65.95 |
| Working hours per week, mean (SE) | 33.19(0.19) |
| Employed | 88.13 |
| Age, mean (SE) | 38.66 (0.20) |
| Smoker | 28.38 |
| Male | 50.45 |
| Degree | 16.07 |
| household income | |
| 1 | 7.32 |
| 2 | 14.59 |
| 3 | 21.21 |
| 4 | 27.65 |
| 5 | 29.21 |
| Health | |
| Morbidity index | 0.82 |
| Morbidity index squared | 0.67 |
| Self reported conditions | |
| Arms | 19.52 |
| Hearing | 4.96 |
| Heart | 7.60 |
| Chest | 10.94 |
| Depression | 5.13 |
| Diabetes | 1.28 |
| Sight | 2.06 |
| Skin | 12.70 |
| Migraine | 8.84 |
| Stomach | 5.27 |
| Other health condition | 4.31 |

*$n$ = 4,796

*Variables expressed as proportion (%) unless otherwise specified

smoking status each of which—based on existing literature—might be expected to influence service use [10, 20]. With respect to, hours worked, income and employment status for example, each might be expected to influence the opportunity cost of time and thereby influence the threshold an individual may apply with respect to ill-health before triggering a healthcare visit. Similarly, education may correlate with health literacy allowing the individual to discern more readily when symptoms warrant investigation and trigger a healthcare visit. Rationales for marital status include risk adversity in shared decision-making [21] and smoking status related to the effects of smoking on health as well as its correlation with other risky behaviours [22]. Details of each are set out in (Tables 1–3). (To avoid issues with the distribution of income while accommodating potential non-linear relationships, the variable was categorised as quintiles. Education was specified as 1 if a degree or higher qualification was attained and zero otherwise consistent with previous studies [10, 20]. Other details are provided in the (Tables 1–3).

**Table 2. Descriptive statistics of the study sample 2004.**

| Variable/parameter | Mean (SE) or Proportion (%) |
|---|---|
| GP Visit | 69.12 |
| Outpatient Visit | 26.26 |
| Married | 65.24 |
| Working hours per week, mean (SE) | 32.72(0.19) |
| Employed | 87.78 |
| Age, mean (SE) | 39.23(0.22) |
| Smoker | 24.78 |
| Male | 49.58 |
| Degree | 18.06 |
| household income | |
| 1 | 5.9 |
| 2 | 11.95 |
| 3 | 19.45 |
| 4 | 29.45 |
| 5 | 33.22 |
| Health | |
| Morbidity | 0.82 |
| Morbidity Square | 0.67 |
| Self reported conditions | |
| Arms | 18.51 |
| Hearing | 4.58 |
| Heart | 9.79 |
| Chest | 10.27 |
| Depression | 5.00 |
| Diabetes | 2.05 |
| Sight | 2.43 |
| Skin | 12.87 |
| Migraine | 7.55 |
| Stomach | 5.70 |
| Other health condition | 4.07 |

*$n$ = 4,416

*Variables expressed as proportion (%) unless otherwise specified

There are limitations associated with the design of surveys for general use. BHPS does not, for example, allow the researcher to readily relate GP visits to specific conditions or to distinguish between GP initiated and user initiated visits. Similarly, in measuring ill health we are obliged to rely on self-reported measures rather than those confirmed by clinical diagnoses. While these limitations exist, they are mitigated by the broader and more detailed range of information available on respondents' circumstances that are contained in these surveys.

## Methods

Descriptive statistics (mean and 95% confidence interval) were used to describe the sample in each year. Bivarate analyses were used to estimate sample sub-groups differences–for example, differences in the morbidity index between distinct types of service user and non-user; between those with higher versus lower income; between those with higher educational attainment and those with less etc. as well as over time. While previous attempts have been made to

**Table 3. Descriptive statistics of the study sample 2008.**

| Variable/parameter | Mean (SE) or Proportion (%) |
|---|---|
| GP Visit | 72.80 |
| Outpatient Visit | 26.09 |
| Married | 67.18 |
| Working hours per week, mean (SE) | 33.06 (0.19) |
| Employed | 91.15 |
| Age, mean (SE) | 40.65 (0.23) |
| Smoker | 22.31 |
| Male | 49.83 |
| Degree | 20.84 |
| household income | |
| 1 | 6.40 |
| 2 | 14.16 |
| 3 | 20.58 |
| 4 | 26.71 |
| 5 | 32.13 |
| Health | |
| Morbidity | 0.78 |
| Morbidity Square | 0.61 |
| Self reported conditions | |
| Arms | 17.55 |
| Hearing | 5.22 |
| Heart | 10.08 |
| Chest | 10.34 |
| Depression | 4.35 |
| Diabetes | 2.85 |
| Sight | 2.77 |
| Skin | 13.59 |
| Migraine | 7.03 |
| Stomach | 5.04 |
| Other health condition | 4.20 |

* $n$ = 3,891

*Variables expressed as proportion (%) unless otherwise specified

incorporate supply into analyses of GP use [10, 20, 23–25] these are vulnerable to the potential for ecological fallacy. In this analysis we chose not to attempt to incorporate supply. Previous studies have attempted to exploit the gatekeeping role of GPs as a way of informing the characterization of health when examining GP use [10]. Here we allow for this using a recursive bivariate probit approach (RBVP) [26]. The RVBP approach allows us to simultaneously exploit the gatekeeping role of the GP with respect to outpatient services to help inform the GP model with respect to the characterization of the respondent's health and to explore the possibility of unobserved heterogeneity in use of services between GP and outpatient service use. The former is achieved by testing the significance, sign and magnitude of the coefficient on outpatient use in the GP function; the latter by testing the sign and significance of correlation in residuals between the two models when estimated jointly. Where a significant positive correlation in errors was detected, this would indicate the omission of characteristics such that where we over(under)-predicted use of GP services we would also over(under)-predict use of

outpatient services. This might arise, for example, where individual risk aversity around health is not incorporated into the function but the worried well are "unduly" likely to visit the GP and "unduly" likely to be referred on to outpatient services. The ability to test for this is potentially informative within a context of changes brought about by QOF and increased investment in the health service where the threshold for onward referral to outpatients may be mutable. Where no correlation in errors is found we revert to the estimation of separate probit models. To compare utilisation in a period where resource constraints were more evident we report mean number of GP and outpatient visits in 2000,2004,2008 and 2015/16, 2016/17, and 2017/ 18. Changes in questionnaire design from the British Household Panel Survey (BHPS) to its successor the "Understanding Society Survey" meant it is not possible to measure health in a consistent fashion across the two surveys. It was therefore not possible to repeat the multivariable analysis for earlier years for 2015 onward. However, the mean figures provide an indication of changes to use as the resource environment changed. Means were based on the class mark for categories that detailed the level of service use reported in the surveys. We use outpatient use to help understand GP use, where outpatient use affectively served as an indication of severity [10]. We do not consider A&E use in this regard given services are not subject to gatekeeping by the GP. We selected the years 2015, 2016, and 2017 because they represent a period during which austerity measures introduced in the early 2010s had accumulated and impacted service utilization. These years provide a clearer understanding of how service use evolved under the influence of constrained public spending. Additionally, taking three consecutive years minimizes the risk of arbitrary selection, as it allows us to observe any emerging trends rather than focusing on a single year that might not be representative of the broader context.

We do not exploit the panel nature of the data but rather examine them as a series of cross sectional analyses. We do this to avoid potential selection effects around attrition across waves of the survey.

## Results

In Tables 1–3, descriptive statistics for each survey year are presented. While changes in the percentage of respondents who used GP and outpatient services respectively increased over time, the increase over the 8-year period was less than 3 percentage points. The slight rise in the count of morbidities–the morbidity index–indicated an increase in sickness over time.

Table 4 compares sub-groups in term of morbidity index over time. BHPS respondents who attended GP services had a higher morbidity index each year compared to respondents who did not consult with a GP. Similarly, respondents who attended outpatients had a higher morbidity index than people who did not attend and their morbidity index was higher compared to GP service users. There was a slight decrease in the morbidity index for several groups over time. Morbidity differences related to education and income were evident–respondents with higher qualifications and those with higher incomes had a lower morbidity index than respondents who were less well educated and those who were less well off. Similarly, differences were evident across age groups those who were older had a higher morbidity index than those who were younger.

Table 5 presents the results of a series of recursive bivariate probits, one for each study year. A significant correlation in the error term was recorded for 2000 and 2004 –suggestive of unobserved heterogeneity in the estimated functions and indicative of the importance of adopting a bivariate estimation approach–this result was not found for 2008 where separate probits were indicated for GP and outpatient services. The positive correlation in errors indicates that where we under-predicted (over-predicted) use of GP services we under-predicted (over-predicted) use of outpatient services (Table 5). The correlation was not statistically

**Table 4. Comparisons of morbidity index (MI) across sub-groups over time.**

| | 2000 | 2004 | 2008 |
|---|---|---|---|
| | **Mean MI (SE)** | **Mean MI (SE)** | **Mean MI (SE)** |
| **GP Visit** | | | |
| Yes | 0.99(0.02) | 1.00(0.02) | 0.95(0.02) |
| No | 0. 43(0.02) | 0.42(0.02) | 0.34(0.02) |
| **Outpatient Visit** | | | |
| Yes | 1.26(0.04) | 1.30(0.04) | 1.19(0.04) |
| No | 0.68(0.01) | 0 .65(0.01) | 0.64(0.01) |
| **Neither GP nor Outpatient user** | 0.40(0.02) | 0.40(0.02) | 0.32(0.02) |
| **Employed** | 0.83(0.01) | 0.83(0.01) | 0.79(0.01) |
| No | 0.76(0.05) | 0.75(0.05) | 0.67(0.05) |
| **Smoker** | 0.89(0.03) | 0.86(0.03) | 0.81(0.03) |
| No | 0.80(0.01) | 0.81(0.02) | 0.78(0.02) |
| **Degree** | 0.73(0.04) | 0.76(0.04) | 0.65(0.03) |
| No | 0.84(0.01) | 0.84(0.02) | 0.82(0.02) |
| **Male** | 0.68(0.02) | 0.71(0.02) | 0.69(0.02) |
| Female | 0.96(0.02) | 0.94(0.02) | 0.88(0.02) |
| **Hours of working per week** | | | |
| 0–22 hours | 1.02(0.04) | 0.89 (0.04) | 0.86(0.04) |
| 23–35 hours | 0.85 (0.03) | 0.85 (0.03) | 0.78(0.03) |
| 36–37 hours | 0.81(0.03) | 0.93(0.04) | 0.82 (0.03) |
| 38–40 hours | 0.69(0.03) | 0.73 (0.03) | 0.74(0.03) |
| >40 hours | 0.68(0.04) | 0.65 (0.04) | 0.69(0.04) |
| **Age** | | | |
| 15–28 | 0.57(0.2) | 0.57(0.02) | 0.50(0.03) |
| 29–40 | 0.779(0.03) | 0.75(0.03) | 0.71(0.03) |
| 41–51 | 0.83(0.03) | 0.84(0.03) | 0.82(0.03) |
| 52–65 | 1.08(0.04) | 1.14(0.04) | 1.06(0.04) |
| 66 and above | 1.57(0.16) | 1.44(0.15) | 1.49(0.19) |
| **Household Income** | | | |
| 1 | 0.90(0.06) | 0.81(0.07) | 0.69(0.06) |
| 2 | 0.90(0.04) | 0.81(0.05) | 0.83(0.05) |
| 3 | 0.83(0.03) | 0.90(0.04) | 0.79(0.03) |
| 4 | 0.84(0.03) | 0.80(0.03) | 0.83(0.03) |
| 5 | 0.74(0.02) | 0.81(0.03) | 0.74(0.02) |
| *n* | 4,796 | 4,416 | 3,891 |

significant in 2008 suggesting separate probits were appropriate to model GP and outpatient use The RBVPs for 2000 and 2004 and the probits for 2008 demonstrated that morbidity (as measured by morbidity index) is positively associated with use of GP and outpatient services (Tables 5, 6). A significant relationship was evident with respect to specific conditions though these should be interpreted with caution given that they also contribute to the morbidity index. Notably, conditions related to sensory impairment–sight and hearing—were consistently and significantly negatively related to use of GP and outpatient services. Socio-economic variables were, generally, not significant in the GP model or the outpatient model, though hours worked was significant in 2008 (Table 6).

Table 7 shows mean of General Practitioner (GP) visits and outpatient visits during periods characterized by relative resource abundance and resource scarcity. As can be seen, the mean

**Table 5.  Recursive bivariate probit analysis.**

|  | 2000 | 20004 | 2008 |
|---|---|---|---|
|  | β (95% C.I) | β (95% C.I) | β (95% C.I) |
| **GP Visit** |  |  |  |
| Outpatient | -0.270 (-0.999, 0.459) | -0.085(-0.968, 0.798) | 0.137(-0.661, 0.935) |
| Married | 0.105(0.001, 0.209) | -0.054(-0.162, 0.053) | 0.001(-0.115, 0.118) |
| Hours | 0.000(-0.003, 0.004) | -0.000(-0.004, 0.004) | -0.000(-0.005, 0.004) |
| Employed | -0.110(-0.255, 0.035) | -0.060(-0.217, 0.096) | 0.045(-0.151, 0.241) |
| Age | -0.268(-0.427, -0.108) | -0.110(-0.285, 0.063) | -0.311(-0.502, -0.119) |
| Smoker | 0.005(-0.091, 0.101) | -0.068(-0.171, 0.035) | -0.085(-0.207, 0.035) |
| Male | -0.497(-0.591, -0.403) | -0.485(-0.581, -0.389) | -0.414(-0.522, -0.306) |
| Degree | -0.017(-0.134, 0.100) | -0.035(-0.152, 0.081) | 0.043(-0.078, 0.165) |
| Morbidity | 0.655(0.354, 0.957) | 0.764(0.394, 1.134) | 0.790(0.449, 1.131) |
| Morbidity square | -0.047(-0.082, -0.013) | -0.025(-0.063, 0.012) | -0.060(-0.105, -0.015) |
| Arms | -0.019(-0.319, 0.281) | -0.312(-0.649, 0.023) | -0.104(-0.462, 0.254) |
| Hearing | -0.372(-0.728, -0.016) | -0.469(-0.839, -0.099) | -0.414(-0.828, 0.000) |
| Heart | 0.126(-0.270, 0.523) | 0.089(-0.316, 0.495) | 0.311(-0.088, 0.710) |
| Chest | -0.192(-0.525, 0.139) | -0.355(-0.742, 0.031) | -0.226(-0.592, 0.138) |
| Depression | 0.298(-0.143, 0.740) | 0.090(-0.368, 0.549) | 0.487(-0.067, 1.042) |
| Diabetes | 0.025(-0.493, 0.544) | -0.053(-0.589, 0.483) | 0.259(-0.295, 0.815) |
| Sight | -0.174(-0.623, 0.274) | -0.717(-1.143, -0.291) | -0.525(-0.963, -0.087) |
| Skin | -0.289(-0.614, 0.034) | -0.400(-0.758, -0.043) | -0.457(-0.814, -0.100) |
| Migraine | -0.321(-0.665, 0.021) | -0.606(-0.995, -0.216) | -0.255(-0.648, 0.137) |
| Stomach | -0.043(-0.393, 0.307) | 0.097(-0.295, 0.491) | 0 omitted |
| Other health problem | 0 omitted | 0 omitted | 0.510(0.193, 0.827) |
| Household Income |  |  |  |
| 2 | 0.014(-0.182, 0.212) | 0.181(-0.049, 0.412) | -0.151(-0.393, 0.090) |
| 3 | -0.057(-0.246, 0.131) | 0.190(-0.027, 0.408) | -0.020(-0.252, 0.211) |
| 4 | -0.077(-0.261, 0.106) | 0.206(-0.007, 0.419) | -0.030(-0.257, 0.195) |
| 5 | 0.017(-0.170, 0.205) | 0.175(-0.036, 0.395) | -0.036(-0.264, 0.191) |
| **Outpatient Visit** |  |  |  |
| Hours | 0.001(-0.002, 0.005) | 0.002(-0.001, 0.006) | 0.005(0.000, 0.010) |
| Married | 0.136(0.027, 0.245) | 0.116(0.007, 0.226) | -0.017(-0.130, 0.095) |
| Employed | -0.005(-0.156, 0.144) | -0.108(-0.261, 0.045) | -0.101(-0.286, 0.083) |
| Age | -0.142(-0.310, 0.025) | 0.094(-0.090, 0.279) | 0.218(0.041, 0.394) |
| Male | -0.236(-0.334, -0.138) | -0.211(-0.309, -0.112) | -0.232(-0.339, -0.125) |
| Smoker | 0.059(-0.040, 0.160) | -0.033(-0.141, 0.075) | 0.095(-0.020, 0.212) |
| Degree | -0.061(-0.186, 0.064) | 0.027(-0.091, 0.146) | -0.053(-0.175, 0.068) |
| Morbidity | 0.903(0.684, 1.121) | 1.019(0.787, 1.251) | 0.705(0.478, 0.932) |
| Morbidity square | 0.001(-0.027, 0.031) | -0.053(-0.078, -0.028) | -0.056(-0.085, -0.026) |
| Arms | -0.432(-0.658, -0.207) | -0.460(-0.702, -0.218) | -0.113(-0.352, 0.125) |
| Hearing | -0.703(-1.001, -0.405) | -0.331(-0.630, -0.329) | -0.465(-0.759, -0.170) |
| Heart | -0.493(-0.756, -0.231) | -0.517(-0.787, -0.248) | -0.304(-0.564, -0.043) |
| Chest | -0.749(-0.988, -0.509) | -0.753(-1.015, -0.490) | -0.428(-0.685, -0.170) |
| Depression | -0.995(-1.282, -0.708) | -0.629(-0.931, -0.328) | -0.276(-0.590, 0.037) |
| Diabetes | 0.170(-0.244, 0.586) | -0.283(-0.668, 0.101) | 0.182(-0.153, 0.519) |
| Sight | -0.989 (-1.345, -0.634) | -0.622(-0.984, -0.261) | -0.408(-0.747, -0.068) |
| Skin | -0.801(-1.041, -0.561) | -0.672(-0.930, -0.414) | -0.374(-0.618, -0.129) |
| Migraine | -0.723(-0.972, -0.475) | -0.849(-1.130, -0.568) | -0.234(-0.508, 0.039) |

**Table 5.** (Continued)

|  | 2000 | 20004 | 2008 |
|---|---|---|---|
|  | β (95% C.I) | β (95% C.I) | β (95% C.I) |
| Stomach | -0.335(-0.606, -0.065) | -0.117(-0.400, 0.164) | 0 omitted |
| Other health problem | 0 omitted | 0 omitted | 0.600(0.376, 0.825) |
| Household Income |  |  |  |
| 2 | -0.126(-0.329, 0.077) | -0.077(-0.322, 0.167) | 0.190(-0.043, 0.424) |
| 3 | -0.074(-0.270, 0.121) | -0.052(-0.281, 0.176) | 0.210(-0.010, 0.431) |
| 4 | -0.050(-0.242, 0.140) | -0.110(-0.332, 0.111) | 0.181(-0.035, 0.398) |
| 5 | 0.047(-0.147, 0.242) | -0.123(-0.349, 0.102) | 0.199(-0.018, 0.416) |
| rho (correlation of residuals) | 0.593(0.128, 1.059) | 0.568(0.018, 1.118) | 0.404(-0.041, 0.850) |
| *n* | 4,796 | 4,416 | 3,891 |

number of GP visits fell by roughly 0.25 visits from (3.002) in 2000 to (2.757) in 2008. Conversely, regarding outpatient visits, the mean number increased by roughly 0.4 visits from (1.186) in 2000 to (1.229) in 2008.

**Table 6.** Probits analysis of GP and outpatient visits.

|  | 2008 |  | 2008 |
|---|---|---|---|
|  | β (95% C.I) |  | β (95% C.I) |
| **GP Visit** |  | **Outpatient Visit** |  |
| Married | -0.001(-0.118, 0.115) | Married | -0.014(-0.128, 0.098) |
| Hours | -0.000(-0.005, 0.004) | Hours | 0.005(0.000, 0.010) |
| Employed | 0.039(-0.154, 0.234) | Employed | -0.095(-0.280, 0.089) |
| Age | -0.304(-0.483, -0.126) | Age | 0.223(0.046, 0.400) |
| Smoker | -0.084(-0.203, 0.034) | Smoker | 0.092(-0.023, 0.209) |
| Male | -0.418(-0.525, -0.310) | Male | -0.228(-0.335, -0.122) |
| Degree | 0.041(-0.079, 0.162) | Degree | -0.054(-0.177, 0.067) |
| Morbidity | 0.830(0.499, 1.161) | Morbidity | 0.701(0.473, 0.929) |
| Morbidity square | -0.056(-0.099, -0.013) | Morbidity square | -0.056(-0.086, -0.026) |
| Arms | -0.134(-0.481, 0.212) | Arms | -0.113(-0.354, 0.126) |
| Hearing | -0.482(-0.889, -0.075) | Hearing | -0.461(-0.756, -0.165) |
| Heart | 0.269(-0.120, 0.658) | Heart | -0.302(-0.564, -0.041) |
| Chest | -0.286(-0.647, 0.075) | Chest | -0.424(-0.682, -0.166) |
| Depression | 0.439(-0.112, 0.990) | Depression | -0.269(-0.582, 0.043) |
| Diabetes | 0.212(-0.331, 0.756) | Diabetes | 0.185(-0.150, 0.522) |
| Sight | -0.580(-1.017, -0.143) | Sight | -0.400(-0.737, -0.062) |
| Skin | -0.490(-0.838, -0.141) | Skin | -0.374(-0.620, -0.129) |
| Migraine | -0.310(-0.704, 0.083) | Migraine | -0.229(-0.503, 0.045) |
| Stomach | 0 omitted | Stomach | Omitted |
| Other health problem | 0.528(0.231, 0.826) | Other health problem | 0.600(0.377, 0.824) |
| Household Income |  | Household Income |  |
| 2 | -0.144(-0.381, 0.093) | 2 | 0.187(-0.047, 0.422) |
| 3 | -0.016(-0.243, 0.210) | 3 | 0.212(-0.009, 0.433) |
| 4 | -0.017(-0.239, 0.204) | 4 | 0.180(-0.038, 0.398) |
| 5 | -0.032(-0.254, 0.189) | 5 | 0.203(-0.013, 0.420) |

* *n* = 3,891

**Table 7. Mean of GP and outpatients visits over time.**

| Year | 2000 | | 2004 | | 2008 | |
|---|---|---|---|---|---|---|
| | Mean (SE) | β (95% C.I) | Mean (SE) | β (95% C.I) | Mean (SE) | β (95% C.I) |
| **GP Visit** | 3.002 | (2.941–3.064) | 2.719 | (2.659–2.779) | 2.757 | (2.696–2.819) |
| *n* | 15,065 | | 14,755 | | 13,440 | |
| **Outpatient Visit** | 1.186 | (1.144–1.229) | 1.190 | (1.147–1.232) | 1.229 | (1.184–1.274) |
| *n* | 15,068 | | 14,757 | | 13,445 | |
| **Year** | **2015–16** | | **2016–17** | | **2017–18** | |
| **GP Visit** | 2.537 | (2.502–2.572) | 2.698 | (2.662–2.733) | 1.195 | (1.168–1.222) |
| *n* | 39,245 | | 37,512 | | 27,496 | |
| **Outpatient Visit** | 1.297 | (1.269–1.325) | 1.310 | (1.282–1.338) | 1.333 | (1.304–1.362) |
| *n* | 39,272 | | 37,522 | | 34,886 | |

In the second period, while the average number of GP visits initially demonstrated an upward trend, ascending from (2.537) in 2015/16 to (2.698) in 2016/17, it declined dramatically to (1.195) in 2017/2018, a fall of more than one full visit. Data on the availability of General Practitioners (GPs) during this period indicate a relatively stable workforce. In 2015, there were 34,492 GPs (FTE), which increased slightly to 34,916 GPs (FTE) in 2017. This suggests that the observed decline in GP visits in 2017/18 is unlikely to be solely attributed to changes in GP availability [27]. In contrast, the mean outpatient visits during the same period displayed slight growth, rising marginally by roughly 0.03 from (1.297) in 2015 to (1.333) in 2018.

## Discussion

The period 2000–2008 witnessed changes to the structure of GP services, the nature of the GP contract and significant investment across the health service generally in Britain. These changes had the potential to produce profound changes in the level and pattern of GP service use and onward referral to outpatient services. However, our results demonstrate that in terms of the percentage of service users across the entire sample, utilization of GP and outpatient services increased only marginally. The morbidity index fell among users of GP and outpatient services. Considered together, these results may be interpreted as providing only weak evidence that sustained investment and an expansion in service use improved access. When considered alongside evidence of increased satisfaction with GP services during this same period [14] the changes may be interpreted as indicative of improvements from the perspective of service users and an expansion in provision of face-to-face consultations. Comparing the two periods 2000–2008 and 2015–2018 shows mean GP utilisation fell more dramatically in second period while outpatient use remained relatively static. This may have reflected the ability of and need for GPs to redirect demand for services elsewhere in the system as the impact of austerity took hold. That is, the decline in demand-led GP use in the second period contrasting with the slight increase in outpatient use suggests that a more constrained resource environment may be indicative of a system under pressure re-balancing how that pressure was managed. While GP utilisation fell slightly during the period of relative resource abundance the fall in use was more dramatic during the period of resource constraint. Outpatient utilisation increased but not significantly during the period of relative resource abundance and a similar pattern was observed for the period of resource constraint. This is supportive of the argument that during the period of relative abundance utilisation of GP services was less constrained than during the period of austerity. While it was not possible to model service use in the same way across the two time periods due to the manner in which health was measured in BHPS compared to Understanding Society, patterns are evident in respect of socio-demographic

characteristics in the earlier period. Further research could extend our analysis to examine impact on use of accident and emergency services.

Socio-economic status remained largely insignificant in terms of its association with use of GP or outpatient services across the earlier period. While socio-demographic characteristics such as age and sex may be associated with differences in service use, factors such as income, education and employment status do not appear to be associated. This differs from results on outpatient and GP consultations found by others using different data for the period 1998–2000 in England [28]. In their study Morris et al found education and aspects of economic activity to significantly predict GP and outpatient use as well as income effecting the likelihood of outpatient use. To what extent these differences in findings relate to differences in the data used, the modelling approach adopted or the time period under examination is unclear. That differences are not evident in respect of income, education and employment status though is potentially reassuring especially with respect to outpatient services (with one exception in 2008 which may be an anomaly) where a greater potential for discrimination may exist [29]. The result regarding the role of sex with respect to GP use is consistent with other studies—men are less likely to use these services. However, the reason why men were as likely as women to use outpatient services is less clear given the gatekeeping role of a GP [22, 30–35].

The results with respect to specific conditions should be interpreted carefully given that each condition contributed to the overall morbidity index score as well as the procedure of entering the function as a dummy variable. It may be noteworthy that individuals with the same level of morbidity who experienced sensory impairment–hearing or vision issues–were less likely to visit a GP and less likely to use outpatient services. These conditions together with skin conditions were the only conditions where individuals were consistently less likely to use GP and outpatient services. It has been reported that individuals may experience hearing loss 10 years before they are referred for assessment and; that of those who consult their GP about hearing only 38% also went to hospital [36]. Barriers to effective use of primary care and secondary care services for those with visual impairment have also been noted in the UK. While this field is complex [37] given that sensory impairment may be correlated with other conditions such as depression and cognitive impairment [38–40], the consistency of results may suggest that even at a time of service investment, people with given conditions may experience differential (including no or very few) benefits from the increased investment. Why this should be the case is unclear and may relate to the availability of privately provided alternatives to those who can afford them. If lower utilization is explained by this, however, what the implications are for equity of access to those who experience need in this area warrants further investigation.

While our analysis links the resource environment to patterns of utilization it is important to also remember the broader context in which services were provided and consumed. Over the period studied, for example, the population increased and aged, with rising multimorbidity contributing to increased needs and the complexity of those needs. Similarly, broader policy changes such as the introduction of the Quality Outcomes Framework changed the incentive structure under which GPs operated, providing financial rewards for the identification and management of chronic conditions such as diabetes. These could each affect patterns of service use and complicate the establishment of causal relationships as does the fact that the data we use does not provide the opportunity for the consistent modelling of utilization. These limitations should be born in mind when reflecting on our findings.

It is unfortunate that the data available did not allow us to model use consistently across periods of distinct relative resource scarcity. They may however provide insights into how service use might change as investment changes that can be generalised. Our analysis suggests that during the period of relative resource abundance not all individuals appear to have

benefitted from this, pockets of what could be unmet needs appearing to persist. During a period of contraction, such as that experienced in the wake of the financial crisis of 2008, where as shown the mean number of visits fell sharply over a relatively short period of time, it follows that some may lose more than others as access to services contracts and harsh decisions around priorities are made. That use of outpatient services by contrast continued to increase–albeit slightly–may as noted be indicative of the system rebalancing to meet needs within a resource constrained environment. As GP services have come under further strain in the wake of the COVID pandemic such differences may have been experienced to an even greater extent. There is a clear need to consider both horizontal and vertical equity during times of contraction and expansion.

There are a number of limitations to our study. Firstly, the data are cross-sectional in nature and, therefore, it is not possible to comment on the causal nature of relationships between use and individual characteristics. Secondly, we did not have scope to incorporate supply explicitly into our analyses. As suggested, variations in access to services related to supply may exist within each year that may contribute to differences in service use. The extent to which supply could be linked to location was limited given the imprecision or area-based measures in the survey which were designed in this way to preserve anonymity. Thirdly, it was not possible to link service use to a particular condition or the severity of that condition. While BHPS affords a rich characterization of a respondent and their health unfortunately this additional information was not contained in it [37]. Fourth, it is not possible to model service use in a consistent fashion using the BHPS successor the Understanding Society Survey. Unfortunately, this transition coincides with a period of austerity where resource constraints became more evident.

## Conclusion

Our study demonstrated that use of GP services is related to distinct periods of relative resource abundance and resource scarcity that may have implications for use of outpatient services. In the period of resource abundance, that those with specific conditions including hearing and vision impairment were consistently less likely to use outpatient services suggests the possibility of ongoing unmet need in respect of certain conditions. As resource scarcity has increased with respect to GP services in particular following the COVID pandemic suggests such patterns of use warrant close examination.

## Author Contributions

**Conceptualization:** Motab Aljohani, Michael Donnelly, Ciaran O'Neill.

**Data curation:** Motab Aljohani, Ciaran O'Neill.

**Formal analysis:** Motab Aljohani.

**Methodology:** Motab Aljohani.

**Supervision:** Ciaran O'Neill.

**Writing – original draft:** Motab Aljohani, Ciaran O'Neill.

**Writing – review & editing:** Motab Aljohani, Michael Donnelly, Ciaran O'Neill.

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
