## [Decision Letter · Decision Letter 0]

28 Apr 2023

PONE-D-22-25682Changing patterns of general practice services during a period of public sector investment in BritainPLOS ONE

Dear Dr. Aljohani,

Thank you for submitting your manuscript to PLOS ONE. After careful consideration, we feel that it has merit but does not fully meet PLOS ONE’s publication criteria as it currently stands. Therefore, we invite you to submit a revised version of the manuscript that addresses the points raised during the review process.

Due to the inavailabilty of addtional reviewers, I have thoroughly reviewed the manuscript and am making this recommendation on the basis of my evaluation and that of Reviewer 1. There are significant concerns regarding the study period such that the study reports on data that are many years out of date, while more recent data are available. It would be useful and indeed necessary to include more recent data in the evaluation, both to provide a stronger basis for inference and to improve the relevance of the results for current policymaking and management praxis. The statistical analysis is suitably conducted and does not require and significant revision, with the exception of an error in Table 1, as noted by Rev. 1. The study appears to be primarily exploratory, but this should be stated more explicitly. There may be a variety of contextual factors that impact the results beyond those mentioned in the manuscript, and the authors may choose to strengthen this aspect of their submission by providing additional evidence from before and after the study period.  The two primary concerns raised by Reviewer 1 are fundamental in nature and should be directly and thoroughly addressed in a revised version of this manuscript.

We look forward to receiving your revised manuscript.

Kind regards,

Blake Byron Walker, Ph.D.

Academic Editor

PLOS ONE

2. Please keep your tables as part of your main manuscript and remove the individual file. Please note that supplementary tables (should remain/ be uploaded) as separate "supporting information" files"

Reviewers' comments:

Reviewer's Responses to Questions

**Comments to the Author**

1. Is the manuscript technically sound, and do the data support the conclusions?

Reviewer #1: Partly

2. Has the statistical analysis been performed appropriately and rigorously? 

Reviewer #1: Yes

3. Have the authors made all data underlying the findings in their manuscript fully available?

Reviewer #1: Yes

4. Is the manuscript presented in an intelligible fashion and written in standard English?

Reviewer #1: Yes

5. Review Comments to the Author

Reviewer #1: This study uses data from the British Household Panel Survey to "examine GP and outpatient service use" and their relationship with different socio-demographic characteristics between 2000 and 2008, a period of organisational reform and increasing expenditure. I will explain two major reservations I have regarding this manuscript, and then make a number of more detailed comments.

While it is true that the period 2000-2008 marked a time of greatly increased health expenditure on the UK NHS, it is not at all clear to me why this analysis should be restricted to this period. Surely also including the periods thereafter (e.g. 2012, 2016), a time of significant austerity (which the authors acknowledge in line 416-418) would provide even deeper insights on underlying relationships (or the lack thereof)? An investigation of putative changes in access related to spending investment would surely be informed equally by periods of low spending increase (or decreases) as by periods of fast growth? Given that this paper has been submitted in 2022, and several waves of BHPS data are available for intervening years, I find this choice hard to explain, and the authors provide no real discussion of why this decision was made.

I am also not really clear as to the fundamental purpose or research question driving this work. It is of course acceptable to undertake a descriptive analysis of relationships between variables, but I am left with the sense that the study was not guided by a clear underlying hypothesis or purpose. As a result, the results and conclusions of this manuscript seem somewhat disjointed and lack a coherent message. In their introduction, the authors cite official activity statistics to show large increases in GP activity in the NHS (but do not attempt to gather similar data for outpatient consultations, which are readily available for the NHS). They state in their discussion (lines 369-373): "our results demonstrate that in terms of the percentage of service users across the entire sample, utilization of GP and outpatient services increased only marginally" and "...only weak evidence that sustained investment and an expansion in service use increased access". They do not discuss the obvious corollary - which is that, while the number of people seeking consultations did not increase dramatically, the frequency of consulations increased amongst those who did (i.e. people who were ill received better access, while people who were not ill still did chose not to seek care). My interpretation of their results is that the % of people seeing a GP did increase; yet - surprisingly given that BHPS provides data on numbers of visits per respondent - they did not attempt to examine whether the frequency of visits per person increased. At the same time, their finding that the morbidity index fell is consistent with improving access - people who were less sick were more likely to have sought care in 2008 than 2000. Yet their Conclusion (lines 435-437) directly contradicts their own Discussion: "Our study demonstrated an increase in utilisation of GP and outpatient services...suggestive of widened access."

Unfortunately, I think this speaks to a limited degree of coherence and consistency underlying this paper. These major concerns unfortunately lead me to make a recommendation of "rejection" for this manuscript. A future revision of this manuscript would benefit not only from including additional years and an analysis of consultation frequency, but would benefit from an improved attention to the context under discussion.

Specific comments:

Line 116-127 - the discussion of QoF etc. is somewhat limited and one-sided: an explicit aim of QoF (better quality chronic disease management) could explicitly be framed (and was at the time) as a means of reducing referrals to secondary (outpatient) care by means of better primary care management. This possibility is not considered. Line 161-166 - this section suggests a misunderstanding which appears throughout the paper (e.g. also lies 391-2). Access to outpatient care is not entirely dependent on access to a GP in the NHS. Most critically, attendance at Accident & Emergency can lead directly to referral to outpatient services, as can a range of other smaller services 9e.g. family planning, STI clinics etc.)

Table 1: How can the "morbidity index squared" be 1.73 when the morbidity index value is 0.82? The square of 0,82 is 0.67

Table 4: I found the presentation of hours of work and Age confusing - these appear to have been split into quintiles, but - especially for age - this seems unusual, and no explanation is given of what age range fits each age quintile.

6. PLOS authors have the option to publish the peer review history of their article (what does this mean?). If published, this will include your full peer review and any attached files.

Reviewer #1: **Yes: **Martin Hensher

---

## [Author Response · Author response to Decision Letter 0]

16 Jun 2023

PLOS ONE

PONE-D-22-25682

Changing patterns of general practice services during a period of public sector investment in Britain

Reviewers' comments:

Reviewer #1: This study uses data from the British Household Panel Survey to "examine GP and outpatient service use" and their relationship with different socio-demographic characteristics between 2000 and 2008, a period of organisational reform and increasing expenditure. I will explain two major reservations I have regarding this manuscript, and then make a number of more detailed comments. 

Reviewer comment:

While it is true that the period 2000-2008 marked a time of greatly increased health expenditure on the UK NHS, it is not at all clear to me why this analysis should be restricted to this period. Surely also including the periods thereafter (e.g., 2012, 2016), a time of significant austerity (which the authors acknowledge in line 416-418) would provide even deeper insights on underlying relationships (or the lack thereof)? An investigation of putative changes in access related to spending investment would surely be informed equally by periods of low spending increase (or decreases) as by periods of fast growth? Given that this paper has been submitted in 2022, and several waves of BHPS data are available for intervening years, I find this choice hard to explain, and the authors provide no real discussion of why this decision was made.

Response:

We thank the reviewer for his comments and suggestions. We have updated our analysis to include data for 2015/16, 2016/17, and 2017/18. Our analysis is limited by changes to the way in which morbidity is recorded after Wave BH18 (2008) with the change over from British Household Panel Survey BHPS to Understanding Society Survey. This had informed our original decision to limit the scope of our analysis. We have, however now included details of the mean number of visits for GP and outpatients for the period before and after the adoption of austerity measures and included discussion of these results. 

More details about the revision of this comment can be found below among other changes inserted to the manuscript.

I am also not really clear as to the fundamental purpose or research question driving this work. It is of course acceptable to undertake a descriptive analysis of relationships between variables, but I am left with the sense that the study was not guided by a clear underlying hypothesis or purpose. As a result, the results and conclusions of this manuscript seem somewhat disjointed and lack a coherent message. In their introduction, the authors cite official activity statistics to show large increases in GP activity in the NHS (but do not attempt to gather similar data for outpatient consultations, which are readily available for the NHS). They state in their discussion (lines 369-373): "our results demonstrate that in terms of the percentage of service users across the entire sample, utilization of GP and outpatient services increased only marginally" and "...only weak evidence that sustained investment and an expansion in service use increased access". They do not discuss the obvious corollary - which is that, while the number of people seeking consultations did not increase dramatically, the frequency of consultations increased amongst those who did (i.e. people who were ill received better access, while people who were not ill still did chose not to seek care). My interpretation of their results is that the % of people seeing a GP did increase; yet - surprisingly given that BHPS provides data on numbers of visits per respondent - they did not attempt to examine whether the frequency of visits per person increased. At the same time, their finding that the morbidity index fell is consistent with improving access - people who were less sick were more likely to have sought care in 2008 than 2000. Yet their Conclusion (lines 435-437) directly contradicts their own Discussion: "Our study demonstrated an increase in utilisation of GP and outpatient services...suggestive of widened access."

Unfortunately, I think this speaks to a limited degree of coherence and consistency underlying this paper. These major concerns unfortunately lead me to make a recommendation of "rejection" for this manuscript. A future revision of this manuscript would benefit not only from including additional years and an analysis of consultation frequency but would benefit from an improved attention to the context under discussion.

Response

We thank the reviewer for his comment, On reflection we agree that the aims of our paper required clarification and our conclusion needed to align more coherently with the discussion. We have therefore redrafted the introduction, method, result discussion, limitation, and conclusions as noted below:

Introduction: we have clarified the study hypothesis as quoted below: 

“In this paper, we analyse three waves of the British Household Panel Survey (BHPS) and of its successor the understanding society survey to examine the relationship between GP and outpatient service use and a changing funding environment.” 

Method: 

“to compare utilisation in a period where resource constraints were more evident, we report mean number of GP and outpatient visits in 2000,2004,2008 and 2015/16, 2016/17, and 2017/18. Changes in questionnaire design from the British Household Panel Survey (BHPS) to its successor the Understanding Society Survey meant it is not possible to measure health in a consistent fashion across the two surveys. It was therefore not possible to repeat the multivariable analysis for earlier years for 2015 onward. However, the mean figures provide an indication of changes to use as the resource environment changed. Means were based on the class mark for categories that detailed the level of service use reported in the surveys.” “We use outpatient use to help understand GP use, where outpatient use affectively served as an indication of severity(10). We do not consider A&E use in this regard given services are not subject to gatekeeping by the GP.”

Result: In results, new tables have been added which report the mean utilisation for the two periods. And the following text: “Table 7 shows mean of General Practitioner (GP) visits and outpatient visits during periods characterized by relative resource abundance and resource scarcity. As can be seen, the mean number of GP visits fell by roughly 0.25 visits from (3.002) in 2000 to (2.757) in 2008. Conversely, regarding outpatient visits, the mean number increased by roughly 0.4 visits from (1.186) in 2000 to (1.229) in 2008.”

In the second period, while the average number of GP visits initially demonstrated an upward trend, ascending from (2.537) in 2015/16 to (2.698) in 2016/17, it declined dramatically to (1.195) in 2017/2018, a fall of more than one full visit. In contrast, the mean outpatient visits during the same period displayed slight growth, rising marginally by roughly 0.03 from (1.297) in 2015 to (1.333) in 2018.

Discussion: 

“Comparing the two periods 2000 – 2008 and 2015 -2018 shows mean GP utilisation fell more dramatically in second period while outpatient use remained relatively static – perhaps reflecting the ability of GPs to re-direct demand for services elsewhere in the system. 

The decline in demand-led GP use in the second period and the virtually unchanged level of outpatient use suggests that a more constrained resource environment may have been an important factor in influencing service use across the two periods. 

“while the GP utilisation fell slightly during the period of relative resource abundance the fall in use was more dramatic during the period of resource constraint. Outpatient utilisation increased but not significantly during the period of relative resource abundance and a similar pattern was observed for the period of resource constraint. This is supportive of the argument that during the period of relative abundance utilisation of GP services was able to increase.” 

While it was not possible to model service use in the same way across the two time periods due to the manner in which health was measured in BHPS compared to Understanding Society, patterns are evident in respect of socio-demographic characteristics in the earlier period.”

“It is unfortunate that the data available did not allow us to model use consistently across periods of distinct relative resource scarcity.”

“where as shown the mean number of visits fell sharply over a relatively short period of time.”

“As GP services have come under further strain in the wake of the COVID pandemic such differences may have been experienced to an even greater extent.”

Limitations:

it is not possible to model service use in a consistent fashion using the BHPS successor the Understanding Society Survey. Unfortunately, this transition coincides with a period of austerity where resource constraints became more evident.”

Conclusion:

“Our study demonstrated that use of GP services is related to distinct periods of relative resource abundance and resource scarcity. In the period of resource abundance, that those with specific conditions including hearing and vision impairment were consistently less likely to use outpatient services suggests the possibility of ongoing unmet need in respect of certain conditions. As resource scarcity has increased with respect to GP services in particular following the COVID pandemic suggests such patterns of use warrant close examination.”

Specific comments:

Line 116-127 - the discussion of QoF etc. is somewhat limited and one-sided: an explicit aim of QoF (better quality chronic disease management) could explicitly be framed (and was at the time) as a means of reducing referrals to secondary (outpatient) care by means of better primary care management. This possibility is not considered. 

Response:

Thank you for drawing our attention to this, we have now extended the discussion to link our finding with implication related to QoF as the following:

“While our analysis links the resource environment to patterns of utilization it is important to also remember the broader context in which services were provided and consumed. Over the period studied, for example, the population increased and aged, with rising multimorbidity contributing to increased needs and the complexity of those needs. Similarly, broader policy changes such as the introduction of the Quality Outcomes Framework changed the incentive structure under which GPs operated, providing financial rewards for the identification and management of chronic conditions such as diabetes. These could each affect patterns of service use and complicate the establishment of causal relationships as does the fact that the data we use does not provide the opportunity for the consistent modelling of utilization. These limitations should be born in mind when reflecting on our findings.”

Line 161-166 - this section suggests a misunderstanding which appears throughout the paper (e.g. also lies 391-2) Access to outpatient care is not entirely dependent on access to a GP in the NHS. Most critically, attendance at Accident & Emergency can lead directly to referral to outpatient services, as can a range of other smaller services 9e.g. family planning, STI clinics etc.)

Response to reviewer:

We thank the reviewer for drawing our attention to this oversight the following text. “We use outpatient use to help understand GP use, where outpatient use effectively serves as an indication of severity. We do not consider A&E use in this regard given services are not subject to gatekeeping by the GP though it is acknowledged that this may provide a route by which some service users may also be referred to outpatient services.”

Table 1: How can the "morbidity index squared" be 1.73 when the morbidity index value is 0.82? The square of 0,82 is 0.67

Response to reviewer: 

We thank the reviewer for drawing our attention to this mistake, it has been corrected.

Table 4: I found the presentation of hours of work and Age confusing - these appear to have been split into quintiles, but - especially for age - this seems unusual, and no explanation is given of what age range fits each age quintile.

Response to reviewer: 

Thank you for your comment, we have added information for the classification of two variables: hours of working and age. 

• For hours of working: Q1 - 0-22 hours; Q2: 23-35 hours; Q3: 36-37 hours; Q4: 38-40 hours; Q5: 41+ hours. 

• For age group: Group (1) 15-28 years old, group (2) 29-40 years old, group (3) 41-51 years old, group (4)52-65, and group (5) 66+ years old, 

We used quintiles to account for the possibility of non-linear associations. We hypothesized that as hours of work or age increased, the likelihood of visiting a healthcare provider might exhibit a different rate of decline or increase. By utilizing quintiles, we aimed to capture these potential variations and provide a more nuanced understanding of the relationship. The use of quintiles allows for the consideration of factors such as the impact of time constraints on healthcare-seeking behaviour and the accumulation of chronic conditions with age. By grouping the data into quintiles, we can identify patterns and trends across distinct sub-groups, enabling a comprehensive analysis of the morbidity index.

---

## [Decision Letter · Decision Letter 1]

7 Dec 2023

PONE-D-22-25682R1Changing patterns of general practice services during a period of public sector investment in BritainPLOS ONE

Dear Dr. Aljohani,

Thank you for submitting your manuscript to PLOS ONE. After careful consideration, we feel that it has merit but does not fully meet PLOS ONE’s publication criteria as it currently stands. Therefore, we invite you to submit a revised version of the manuscript that addresses the points raised during the review process.

The previous reviewer has assessed the changes, and have provided some additional comments below. Please consider their suggestions for framing the findings. Please also note that making the data available, or updating the Data availability statement to indicate where others may find any third party data, is a requirement for publication in PLOS ONE.

We look forward to receiving your revised manuscript.

Kind regards,

Hanna Landenmark

Staff Editor

PLOS ONE

Journal Requirements:

Reviewers' comments:

Reviewer's Responses to Questions

**Comments to the Author**

1. If the authors have adequately addressed your comments raised in a previous round of review and you feel that this manuscript is now acceptable for publication, you may indicate that here to bypass the “Comments to the Author” section, enter your conflict of interest statement in the “Confidential to Editor” section, and submit your "Accept" recommendation.

Reviewer #1: (No Response)

2. Is the manuscript technically sound, and do the data support the conclusions?

Reviewer #1: Partly

3. Has the statistical analysis been performed appropriately and rigorously? 

Reviewer #1: Yes

4. Have the authors made all data underlying the findings in their manuscript fully available?

Reviewer #1: No

5. Is the manuscript presented in an intelligible fashion and written in standard English?

Reviewer #1: Yes

6. Review Comments to the Author

Reviewer #1: Thank you for the efforts that the authors have gone to in addressing my original comments. The paper is greatly improved by the addition of later data, notwithstanding the limitations of the new version of the survey. I would strongly encourage the authors to spend a little more time on their Discussion and Conclusions sections however - I find these a little thin and not really bringing out all of the key issues. What does it mean that outpatient utilisation consistently increased while GP utilisation fell over time? It means that relative accessibility of primary care is decreasing, and use of hospital-based ambulatory care is increasing. This is a critical finding - because it shows that reality has been the exact opposite of decades of policy efforts seeking to decrease secondary care use by strengthening primary care. I would like to see these deeper policy implications discussed rather more directly. It was also not clear to me whether the authors are making their data available online or in a supplement - this should be addressed before acceptance.

7. PLOS authors have the option to publish the peer review history of their article (what does this mean?). If published, this will include your full peer review and any attached files.

Reviewer #1: **Yes: **Martin Hensher

---

## [Author Response · Author response to Decision Letter 1]

22 Jan 2024

Reviewers' comments:

Reviewer #1: Thank you for the efforts that the authors have gone to in addressing my original comments. The paper is greatly improved by the addition of later data, notwithstanding the limitations of the new version of the survey. I would strongly encourage the authors to spend a little more time on their Discussion and Conclusions sections however - I find these a little thin and not really bringing out all of the key issues. What does it mean that outpatient utilisation consistently increased while GP utilisation fell over time? It means that relative accessibility of primary care is decreasing, and use of hospital-based ambulatory care is increasing. This is a critical finding - because it shows that reality has been the exact opposite of decades of policy efforts seeking to decrease secondary care use by strengthening primary care. I would like to see these deeper policy implications discussed rather more directly. It was also not clear to me whether the authors are making their data available online or in a supplement - this should be addressed before acceptance.

Response:

Thank you for your valuable feedback and for acknowledging the improvements made to the paper. We appreciate your suggestion to further strengthen the Discussion and Conclusions sections and have added additional text to these. 

Regarding the availability of used data. Data from the British Household Panel Survey were collected by the Institute for Social and Economic Research at the University of Essex and made available by the UK Data Archive. We have added a note that the data can be located from this source.

---

## [Decision Letter · Decision Letter 2]

22 Apr 2024

PONE-D-22-25682R2Changing patterns of general practice services during a period of public sector investment in BritainPLOS ONE

Dear Dr. Aljohani,

Thank you for submitting your manuscript to PLOS ONE. After careful consideration, we feel that it has merit but does not fully meet PLOS ONE’s publication criteria as it currently stands. Therefore, we invite you to submit a revised version of the manuscript that addresses the points raised during the review process.

We look forward to receiving your revised manuscript.

Kind regards,

André Luis C Ramalho, PhD

Academic Editor

PLOS ONE

Journal Requirements:

Reviewers' comments:

Reviewer's Responses to Questions

**Comments to the Author**

1. If the authors have adequately addressed your comments raised in a previous round of review and you feel that this manuscript is now acceptable for publication, you may indicate that here to bypass the “Comments to the Author” section, enter your conflict of interest statement in the “Confidential to Editor” section, and submit your "Accept" recommendation.

Reviewer #1: All comments have been addressed

Reviewer #2: (No Response)

2. Is the manuscript technically sound, and do the data support the conclusions?

Reviewer #1: Yes

Reviewer #2: Yes

3. Has the statistical analysis been performed appropriately and rigorously? 

Reviewer #1: Yes

Reviewer #2: I Don't Know

4. Have the authors made all data underlying the findings in their manuscript fully available?

Reviewer #1: Yes

Reviewer #2: Yes

5. Is the manuscript presented in an intelligible fashion and written in standard English?

Reviewer #1: Yes

Reviewer #2: Yes

6. Review Comments to the Author

Reviewer #1: (No Response)

Reviewer #2: This study presents a compelling analysis of the shifting patterns in service utilization correlating with variations in investment levels. However, the lack of a standardized population for comparison undermines the strength of these findings.

Additionally, the rationale for selecting data from the years 2015, 2016, and 2017 remains unexplained. It would enhance the study's relevance if the authors could present data on annual spending for primary care services rather than relying on broader economic indicators like GDP healthcare expenditure.

Furthermore, this information is accessible in the NHS's annual reports. To improve the study's comprehensiveness and accuracy, the authors are encouraged to include:

The number of general practitioners available during these specified years.

The population size for each of these years.

7. PLOS authors have the option to publish the peer review history of their article (what does this mean?). If published, this will include your full peer review and any attached files.

Reviewer #1: **Yes: **Prof. Martin Hensher

Reviewer #2: **Yes: **Ang Yee Gary

---

## [Author Response · Author response to Decision Letter 2]

16 Oct 2024

Response to reviewer comments: 

Reviewer comment:

This study presents a compelling analysis of the shifting patterns in service utilization correlating with variations in investment levels. However, the lack of a standardized population for comparison undermines the strength of these findings. Additionally, the rationale for selecting data from the years 2015, 2016, and 2017 remains unexplained. 

Response: 

We thank the reviewer for his comments. Our analysis is limited by changes to the way in which morbidity is recorded after Wave BH18 (2008) with the change over from British Household Panel Survey BHPS to Understanding Society Survey. This had informed our original decision to limit the scope of our analysis. We have, however included details of the mean number of visits for GP and outpatients for the period before and after the adoption of austerity measures and included discussion of these results. 

This explanation is included in the manuscript in lines 219–229 of the discussion section:

“To compare utilisation in a period where resource constraints were more evident we report mean number of GP and outpatient visits in 2000,2004,2008 and 2015/16, 2016/17, and 2017/18. Changes in questionnaire design from the British Household Panel Survey (BHPS) to its successor the “Understanding Society Survey” meant it is not possible to measure health in a consistent fashion across the two surveys. It was therefore not possible to repeat the multivariable analysis for earlier years for 2015 onward. However, the mean figures provide an indication of changes to use as the resource environment changed. Means were based on the class mark for categories that detailed the level of service use reported in the surveys. We use outpatient use to help understand GP use, where outpatient use affectively served as an indication of severity (10). We do not consider A&E use in this regard given services are not subject to gatekeeping by the GP”.

In lines 424–428 of the discussion section:

“While it was not possible to model service use in the same way across the two time periods due to the manner in which health was measured in BHPS compared to Understanding Society, patterns are evident in respect of socio-demographic characteristics in the earlier period. Further research could extend our analysis to examine the impact on use of accident and emergency services.”

Additionally, this point is reiterated in lines 497–500:

“It is not possible to model service use in a consistent fashion using the BHPS successor the Understanding Society Survey. Unfortunately, this transition coincides with a period of austerity where resource constraints became more evident.”

Additional text has been added in the method section: 

 “We selected the years 2015, 2016, and 2017 because they represent a period during which austerity measures introduced in the early 2010s had accumulated and impacted service utilization. These years provide a clearer understanding of how service use evolved under the influence of constrained public spending. Additionally, taking three consecutive years minimizes the risk of arbitrary selection, as it allows us to observe any emerging trends rather than focusing on a single year that might not be representative of the broader context”.

Reviewer comment:

 To improve the study's comprehensiveness and accuracy, the authors are encouraged to include: The number of general practitioners available during these specified years.

The population size for each of these years.

Reponse: 

We have added the requested information in the results section. The additional text reads:

“Data on the availability of General Practitioners (GPs) during this period indicate a relatively stable workforce. In 2015, there were 34,492 GPs (FTE), which increased slightly to 34,916 GPs (FTE) in 2017. This suggests that the observed decline in GP visits in 2017/18 is unlikely to be solely attributed to changes in GP availability”.

---

## [Editor Report · Decision Letter 3]

21 Oct 2024

Changing patterns of general practice services during a period of public sector investment in Britain

PONE-D-22-25682R3

Dear Dr. Aljohani,

We’re pleased to inform you that your manuscript has been judged scientifically suitable for publication and will be formally accepted for publication once it meets all outstanding technical requirements.

The revisions made by the authors have successfully addressed the minor changes suggested by the reviewer after editorial review, and no further round of review is necessary.

Kind regards,

André Ramalho, PhD

Academic Editor

PLOS ONE

---

## [Editor Report · Acceptance letter]

6 Nov 2024

PONE-D-22-25682R3 

PLOS ONE

Dear Dr. Aljohani, 

I'm pleased to inform you that your manuscript has been deemed suitable for publication in PLOS ONE. Congratulations! Your manuscript is now being handed over to our production team.

Kind regards, 

on behalf of

Prof. Dr. André Ramalho 

Academic Editor

PLOS ONE